# Cardiac Progenitor Cell Exosomal miR-935 Protects against Oxidative Stress

**DOI:** 10.3390/cells12182300

**Published:** 2023-09-18

**Authors:** Susana Aguilar, Paula García-Olloqui, Lidia Amigo-Morán, José Luis Torán, Juan Antonio López, Guillermo Albericio, Gloria Abizanda, Diego Herrero, África Vales, Saray Rodríguez-Diaz, Marina Higuera, Rubén García-Martín, Jesús Vázquez, Carmen Mora, Gloria González-Aseguinolaza, Felipe Prosper, Beatriz Pelacho, Antonio Bernad

**Affiliations:** 1Cardiac Stem Cells Lab, Centro Nacional de Biotecnología (CNB-CSIC), Department of Immunology and Oncology, Campus Universidad Autónoma de Madrid, 28049 Madrid, Spain; susanaaguilar@guardiacivil.es (S.A.); lidia.amigo@uc3m.es (L.A.-M.); jltoran@cnb.csic.es (J.L.T.); galbericio@cnb.csic.es (G.A.); d.herrero@ictan.csic.es (D.H.); mhiguera@cnb.csic.es (M.H.); ruben.garcia@cnb.csic.es (R.G.-M.); carmen.mora@medsir.org (C.M.); 2Center for Applied Medical Research (CIMA), Regenerative Medicine Department, University of Navarra, 31008 Pamplona, Spain; pgarciao@unav.es (P.G.-O.); gabizanda@unav.es (G.A.); avales@unav.es (Á.V.); srodrigu@unav.es (S.R.-D.); fprosper@unav.es (F.P.); 3Instituto de Investigación Sanitaria de Navarra (IdiSNA), 31008 Pamplona, Spain; ggasegui@unav.es; 4Cardiovascular Proteomics Laboratory, Spanish National Cardiovascular Research Center (CNIC), Melchor Fernández Almagro 3, 28029 Madrid, Spain; jalopez@cnic.es (J.A.L.); jvazquez@cnic.es (J.V.); 5CIBER de Enfermedades Cardiovasculares (CIBERCV), 28029 Madrid, Spain; 6Section of Integrative Physiology and Metabolism, Joslin Diabetes Center, Harvard Medical School, Boston, MA 02215, USA; 7Program of Gene Therapy, Center for Applied Medical Research (CIMA), University of Navarra, 31008 Pamplona, Spain; 8Department of Hematology and Cell Therapy, Clínica Universidad de Navarra, 30008 Pamplona, Spain

**Keywords:** cardiac progenitor cell, CPC, exosome, mi-RNA, miR-935, oxidative stress, apoptosis, antagomiR

## Abstract

Oxidative stress-induced myocardial apoptosis and necrosis are critically involved in ischemic infarction, and several sources of extracellular vesicles appear to be enriched in therapeutic activities. The central objective was to identify and validate the differential exosome miRNA repertoire in human cardiac progenitor cells (CPC). CPC exosomes were first analyzed by LC-MS/MS and compared by RNAseq with exomes of human mesenchymal stromal cells and human fibroblasts to define their differential exosome miRNA repertoire (exo-miR^SEL^). Proteomics demonstrated a highly significant representation of cardiovascular development functions and angiogenesis in CPC exosomes, and RNAseq analysis yielded about 350 different miRNAs; among the exo-miR^SEL^ population, miR-935 was confirmed as the miRNA most significantly up-regulated; interestingly, miR-935 was also found to be preferentially expressed in mouse primary cardiac Bmi1^+high^ CPC, a population highly enriched in progenitors. Furthermore, it was found that transfection of an miR-935 antagomiR combined with oxidative stress treatment provoked a significant increment both in apoptotic and necrotic populations, whereas transfection of a miR-935 mimic did not modify the response. Conclusion. miR-935 is a highly differentially expressed miRNA in exo-miR^SEL^, and its expression reduction promotes oxidative stress-associated apoptosis. MiR-935, together with other exosomal miRNA members, could counteract oxidative stress-related apoptosis, at least in CPC surroundings.

## 1. Introduction

Stem cell therapy is being developed to promote repair and eventually regenerate the damaged heart, but currently its putative therapeutic effect is hampered by poor engraftment/survival of transplanted cells [1,2]. Cardiosphere-derived cells (CDC) and cardiac progenitor cells (CPC) have proven in preclinical studies to induce functional improvements in myocardial infarction (MI) disease [3,4,5,6]. Phase I human trials have been conducted with autologous CPC/CDC in both ischemic and non-ischemic clinical scenarios [7,8]. Moreover, recently, a phase 1 safety cohort of pediatric patients (<18 years old) with dilated cardiomyopathy, have been treated with autologous CDC; after one-year follow-up, some functional improvement has been demonstrated [9]. However, clinical evaluation of allogeneic (CPC/CDC) cardiac-derived cell products yielded unsatisfactory preliminary efficacy results [10,11], suggesting that further improvements are required to develop efficient cellular medicines, probably paralleled by a deeper comprehension of heart homeostasis and response to damage.

It is accepted that a substantial part of the CDC/CPC repair effects are mediated by secreted paracrine factors [12]. These factors are postulated to promote cell-at-risk survival, protect against myocardial ischemia, stimulate endogenous repair, and result in durable benefits [8,13]. Consistent with this hypothesis, a recent functional analysis in mice concluded that several cell-based therapies improve heart function after ischemia-reperfusion injury through an acute sterile immune response of wound healing [14]. A complementary source for regulatory factors are extracellular vesicles (EV) that have been evaluated to minimize cardiac injury (reviewed in [15]). Now it is accepted that EV populations (microvesicles and exosomes) correlate with physiological cell status and could critically participate both in non-disease contexts, such as moderate exercise [16] and in pathology genesis and/or evolution [17], mediating tissue cross-talk and long-distance effects [18]. Therefore, exosomes are being evaluated both as disease markers and as a promising strategy for tissue regeneration (reviewed in [19]). An interesting example is the combined intramyocardial administration of EV, followed by several doses of mesenchymal stem cells (MSC), that demonstrated to substantially improve heart function after acute MI (AMI); this approach improved cardiac function, reduced infarct size and apoptosis, and promoted neovascularization [20].

Each exosome source demonstrates a different composition, but probably the most relevant source of regulatory signaling is associated with their RNA content, including mRNA, microRNA (miRNA), and long non-coding RNA [21]. In particular, most efforts in cardiovascular therapy have been focused on the use of exosomal particles derived from CPC/CDC and MSC, rendering some promising preliminary results (reviewed in [22]). In vitro, CPC exosomes prevent staurosporine-induced cardiomyocyte (CM) apoptosis more effectively than bone marrow MSC exosomes [23]. In addition, human CDC exosomes, delivered intramyocardially, decreased scarring, halted adverse remodeling, and improved Left Ventricle Ejection Fraction (LVEF) in acute and chronic porcine MI [24]. Exosomes, therefore, are considered to play a major role in cardiovascular regulation (reviewed in [19]).

CPC- and CDC-derived exosomes are enriched in some cardioprotective miRNAs, such as miR-146a-3p (reviewed in [25]). A large proportion of exosome-enriched miRNAs, especially those described in cardiovascular disease and wound healing, have also been mechanistically characterized, both in in vitro models and in rodents. The more advanced targets have also been evaluated in large animals. One of the first studies that investigated the use of therapeutic miRNA in large animal models was focused on miR-92a [26] which is currently being evaluated in its early clinical phases, trying to exploit its proangiogenic properties. Six additional miRNAs have also obtained relevant preclinical results in pigs and rabbits, and miR-132 is currently under clinical evaluation [27]. Finally, interference with miRNA expression strategies is being evaluated using antimiR, shRNA, GapmeR, or adeno-associated vectors (AAV) [26].

Here, we have thoroughly analyzed the preferential proteins (LC-MS/MS) and miRNA exosomal repertoire of human CPC. We have identified miR-935 as a highly differentially expressed miRNA in CPC compared with other reference cell lineages; in agreement, miR-935 is also preferentially represented in murine CPC. Moreover, its functional evaluation demonstrated the induction of oxidative stress-associated apoptosis after its inhibition. Hence, miR-935, combined with other exosomal miRNAs, could act together to counteract oxidative stress-related apoptosis in different contexts.

## 2. Materials and Methods

### 2.1. Cells and Culture Conditions

#### 2.1.1. Primary Cells

hCPC were obtained from human right atria appendages from adult donors with no relevant cardiac pathology and subjected to cardiac surgery with extracorporeal circulation; during the procedure, this tissue is normally discarded during cannulation. hCPC were isolated from human myocardial samples by c-kit immunoselection, as described [28]. CPC1-CPC-3, used throughout the whole manuscript, correspond to representative isolates obtained from independent human samples and have been previously described [28,29,30]. hCPC were maintained and expanded as previously indicated, essentially under equivalent conditions to those used in the CAREMI clinical trial (EudraCT 2013–001358–81). All cells were expanded and manipulated (induction of oxidative damage and transfections) in a 3% O_2_/5% CO_2_ atmosphere, which mimics closer physiologic conditions and reduces the senescence evolution of the cultures [29,30,31]. More details can be found in the Appendix A [expanded methods].

Primary Sca1^+^Bmi1^+^CD45^−^population (B-CPC) were sorted from Tamoxifen (Tx)-Induced Bmi1CreERT/+/Rosa26-floxed-YFP animals [32,33]. For details, see Appendix A [expanded methods]. Cardiomyocytes (CM), non-myocyte cardiac cells (Non-CM), and B-CPC were obtained simultaneously from each heart; B-CPC were cultured as previously described [32]; for details, see Appendix A [expanded methods].

#### 2.1.2. Cell Lines

All cell lines used and their specific culture conditions can be found in the Appendix A [expanded methods]. B-CPC^IMM^ is an SV40-immortalized cell line derived from B-CPC described in Figure 1; it was maintained in equivalent conditions to B-CPC.

When oxidative treatment was induced, H_2_O_2_ (Thermo Fisher Scientific, Madrid, Spain) was freshly prepared at the indicated doses (100 mM–20 mM) and added to the different cultures or co-cultures for 24–48 h. Cells were then recovered and evaluated for the different parameters shown.

HL-1 (a gift from Dr. Fernández-Avilés (HUGM, Madrid, Spain)) and HUVEC (a gift from Dr. Santos Mañes, CNB, Madrid, Spain) were cultured at 37 °C, 21% O_2_, 5% CO_2_, in the indicated conditions (see Appendix A [expanded methods]).

### 2.2. Exosome Isolation and Characterization

Exosomes were obtained by differential centrifugation of the medium derived from semiconfluent cultures of the different primary cells and lines [28], as previously described [34]. Briefly, cell cultures were maintained, as previously indicated, until 80–90% confluence; cells were then washed three times with PBS, and the medium was changed to 20 mL serum-free DMEM/per 225 cm^2^ cell culture flask. Cell culture supernatant (160 mL) was harvested at 24 h and centrifuged at 300× *g* (1290 rpm) for 10 min at 4 °C to pellet dead cells and bulky debris (Eppendorf Centrifuge 5804 R, A-4–44 swinging-bucket rotor), followed by centrifugation at 2000× *g* (3333 rpm) for 10 min at 4 °C to eliminate debris and large vesicles. The supernatant was then centrifuged at 10,000× *g* for 30 min in a Type 60 Ti fixed-angle rotor at 4 °C. After another centrifugation in a Type 60 Ti rotor (100,000× *g*) for 70 min at 4 °C, the supernatant was collected and subjected to 100,000× *g* centrifugation in an Optima L-100 XP ultracentrifuge (Beckman Coulter). The final 100,000-g pellet was suspended in cold PBS in 1/10 of the initial volume of supernatant and subjected to test centrifugations at 10,000× *g*. The final supernatant was collected and characterized (electron microscopy analysis, mean population size, and particle concentration) (Figure 2) prior to aliquotation (50–100 uL) and –80 °C conservation. Well-established exosome markers (CD63, CD9, HSP70, and CD81) were evaluated in CPC1-CPC3 independent isolates by Western blot (Figure 2D). Bradford analysis was performed for an estimation of protein concentration and Nanosight (Malvern Panalytical Ltd., Malvern, UK) for mean particle size (Figure 2C); regular concentrations were 2–3 × 10^11^ particles/mL with a mean size of 180 nm. For microscopy characterization, a JEOL-JEM 1011 transmission electron microscope (JEOL USA, Inc., Peabody, MA, USA) was used.

### 2.3. Label-Free Proteomics and Bioinformatics Analysis

Exosomal fractions were obtained from CPC (CPC1-3), MSC (19, 33, 45), and HDF (F1, F2, and F3) isolates. For protein extract preparation, pellets were resuspended in lysis buffer (50 mM Tris-HCl pH 8.5, 4% sodium dodecyl sulfate (SDS), and 50 mM dithiothreitol (DTT)), boiled for 5 min, and incubated for 30 min at room temperature for full protein solubilization. Total proteins (~200 µg) were trypsinized, and the resulting tryptic peptides were dissolved in 0.1% formic acid (FA) and loaded into an nLC-MS/MS system. To increase proteome coverage, tryptic peptides were fractionated by cation exchange chromatography (Oasis HLB-MCX columns). Mass spectrometry proteomics data were deposited in the Peptide Atlas and are accessible through the PASS00827 accession number.

For systems biology and bioinformatic analysis, proteins were grouped into functional categories from a database created by joining categories and pathways from Gene Ontology, Reactome, PIR, and KEGG, as well as Ingenuity Pathways Analysis databases (www.ingenuity.com; versions 12 August 2014).

### 2.4. Exosomal miRNA RNAseq and System Biology Analysis

miRNA were purified from CPC (CPC1-3), MSC (19, 33), and HDF (F1, F2) exosomes, as described [29]. miRNAseq libraries were obtained using the TruSeq RNA Sample Preparation v2 Kit (Illumina 1.9) and sequenced following the standard miRNA sequencing protocol. The quality, quantity, and size distribution of the Illumina libraries were determined using the DNA-1000 Kit (Agilent Bioanalyzer, Madrid, Spain). Libraries were sequenced on the Genome Analyzer IIx following the standard miRNA sequencing protocol with the TruSeq SBS Kit v5. Fastq files containing reads for each library were extracted and demultiplexed using the Casava v1.8.2 pipeline. Sequenced reads were quality-controlled and pre-processed using cutadapt v1.6 to remove adaptor contaminants as described previously [30]. Resulting reads were aligned and gene expression quantified using RSEM v1.1.19 over human reference GRCh37 and Ensembl genebuild 65. Only genes with at least 1 count per million in at least 3 samples were considered for statistical analysis. The data were then normalized and deferential expression tested using the bioconductor package EdgeR v3.0.8. Differentially expressed miRNAs were considered those with a Benjamini-Hochberg adjusted *p* value ≤ 0.05. For the set of differentially expressed genes, functional analysis was performed using the topGO v2.10 Bioconductor R package [30] with annotations from org.Hs.eg.db and GO.db v2.8. For functional analysis, GO annotations were used. For systems biology analysis, miRNA were grouped into functional categories from the Gene Ontology and Ingenuity Pathways Analysis databases (www.ingenuity.com; versions 12 August 2014). All transcriptomic data from this study are deposited in the Gene Expression Omnibus (GEO) database and are accessible through the GSE84070 accession number. For gene enrichment analysis, hsa-miR-935 and mmu-miR-935 putative target genes were selected from TargetScan http://www.targetscan.org/vert_72/, accessed on 1 August 2023) and the miRbase databases. Gene Ontology (GO) enrichment was performed using the enrichGO function from the clusterProfiler R package [35] and selected those categories with an enrichment *p* value < 0.05.

### 2.5. Angiogenic Activity Determination

It was carried out as previously described [29]. In brief, human umbilical vein endothelial cells (HUVEC) were seeded (5 × 10^4^ cells/cm^2^) on Matrigel (356234, BD Biosciences, San Jose, CA, USA). Cells were incubated with CPC exosome fractions (10^7^ or 10^8^, 6 h) in serum-free medium 199 (Biowest; Nuaille, France) (negative control) or medium 199 supplemented with 2% fetal bovine serum (FBS) + 3 ng/mL FGF or 5 ng/mL VEGF (positive control). Cultures were evaluated 48h later and images were analyzed with ImageJ [M1] software (US National Institutes of Health, Bethesda, MD, USA).

### 2.6. Co-Cultures Experiments

B-CPC^IMM^ or CPC-derived exosomes (10^9^ exosomal particles) were co-cultured with target cells (HL-1, HDF, and HUVEC; 6–48 h; 37 °C, 21% O_2_/5% CO_2_) on subconfluent cultures (seeded at 4 × 10^4^ cells/cm^2^ on 0.1% gelatin-treated plates) and compared with appropriate controls. B-CPC^IMM^-conditioned medium was obtained as previously described [30], using low (3%) oxygen-cultured (48 h) cells in IMDM. In all cases, co-cultures were maintained in the B-CPC culture medium. When indicated, those co-cultures were prolonged with the addition of H_2_O_2_ (100 mM) for an additional 24–48 h. B-CPC^IMM^ are YFP+, so they were sorted out and analyzed. Evaluation of apoptosis and necrosis cells was estimated by propidium iodide and annexin staining, followed by cell cytometry (see below).

### 2.7. Flow Cytometry

Cells were detached with trypsin-EDTA, washed with PBS and 0.01% BSA, and incubated with the primary antibody for 1 h on ice. After extensive washing, cells were incubated with fluorescent secondary antibodies (30 min on ice), washed with PBS + 0.01% BSA, and analyzed on a FACS Canto 3L flow cytometer (BD Biosciences). The proliferation rate in mouse cardiac fibroblasts (mCF) and HL-1 was evaluated by Edu or BrdU incorporation, as described below. When indicated, cells were previously transfected with miR-935 or the antigomiR (INH) and compared with cells transfected with the corresponding controls (scramble) or non-transfected controls. The estimation of apoptotic (Annexin^+^/PI^−^) and necrotic (Annexin^+^/PI^+^) cells was evaluated by flow cytometry analysis, as previously described [36]. Histograms indicate the percentage of proliferating cells in each condition. For flow cytometry analysis, samples were analyzed with a Beckman Coulter Moflow XDP cell sorter, a Beckman Coulter GALLIOS analyzer, BD FACSCanto II and FACSAria III (BD Biosciences) cytometers. FlowJo vX1 (TreeStar), version 10.5.3, and FlowLogic software, version 8.7, (Inivai Technologies) were used for data analysis. B-CPC and other cardiac subpopulations, both in homeostasis and after AMI, were isolated as previously described [37].

### 2.8. Viability, Proliferation and Apoptosis Assays

To evaluate cell viability, they were detached with trypsin-EDTA 48 h post-transfection, labeled with DAPI (1/1000; Sigma-Aldrich), and quantified by flow cytometry on a FACS Canto 3L flow cytometer (BD Biosciences). Proliferation assays (Edu/BrdU incorporation) were carried out essentially as previously described [36].

For apoptosis analysis, cells were exposed to H_2_O_2_ (500 mM) for 5 h [38,39], then collected (including detached cells) and labeled at 4 °C for 15 min with AnnexinV-FITC (diluted 1:10) in the binding buffer provided by the manufacturer (ApoScreen^®^ Annexin V Apoptosis Kit-FITC; Southern Biotech, Birmingham, AL, USA). Labeled cells were washed with PBS/0.01% BSA and resuspended in 390 μL of binding buffer. Propidium iodide (50 mg/mL, Beckman Couler, Nyon, Switzerland) was added (1:40 dilution) for dual staining, and cells were analyzed by flow cytometry. DAPI and AnnexinV/PI-positive cells were quantified on a FACS Canto 3L flow cytometer (BD Biosciences).

Finally, apoptosis evaluation was alternatively carried out using the Proteome Profiler Mouse Apoptosis Array Kit (R&D Systems; Minneapolis, MN, USA) and following the manufacturer’s indications. Briefly, HL-1 cells were transfected with INH (25 nM) and subjected to oxidative stress (H_2_O_2_ 100 mM, during 24 h) [38,39], in comparison with control HL-1 cells (transfected with the scramble construct (25 nM) and also subjected to oxidative damage). Cells were recovered, lysed, and used for each array’s membrane-based sandwich immunoassay. Capture antibodies spotted in duplicate on nitrocellulose membranes bind to specific target proteins present in the sample.

### 2.9. Gene Silencing Assays

Indicated cell lines were transfected in Opti-MEM medium (Gibco, Invitrogen, Madrid, Spain) with 10–60 nM of miR-935 mimic (mimic) or small interfering RNA against miR-935 (INH) and a control sequence (scramble; C). Mmu-miR-935 (mirVana^®^ miRNA-935 mimic), as well as its respective inhibitor (mirVana^®^ miRNA-935 inhibitor) and appropriate controls (mirVana^®^ miRNA Mimic Negative Control #1 and mirVana^®^ miRNA Inhibitor Negative Control #1), were provided by ThermoFisher Scientific (Appendix A). They were first titrated in HL-1 cells and reconfirmed for each cell lineage evaluated. All cells were transfected using Lipofectamine 2000 Reagent (Invitrogen, Thermo Fisher Scientific); cells were maintained overnight with the transfection mix, then washed and cultured in their corresponding medium for an additional 24–48 h. RT-qPCR was used to check silencing efficiency. Functional effects were mainly tested 48 h post-transfection, as maximum inhibition efficiency was confirmed at this time point for most cell lines. Samples were kept frozen at –80 °C.

### 2.10. miRNA Vector Cloning and AAV Viral Vector Production

The miRNA constructs were created by redesigning the well-studied human miR-30 [40]. The stem of the primary miR-30 transcript was replaced with a specific duplex sequence of miRNA-935, or a scramble sequence (obtained from Sh Scramble hairpin plKO viral vector Addgene #1864, Watertown, MA, USA), and these sequences were cloned into pUC57 (Genscript, Leiden, Netherlands). Both plasmids (pUC57-miR935 and pUC57-Scramble) were digested and purified fragments subcloned into digested AAV-TnT-LUC-WPRE-pA [41] to replace the luciferase gene by miRNA-935 or scramble sequences. AAV vector batches were produced as previously described [41].

### 2.11. RT-qPCR Analyses

Total RNA from cells and digested tissue pellets was extracted using the TRIzol™ Reagent (Thermo Fisher Scientific) and isolated according to the manufacturer’s instructions. Target genes were evaluated by quantitative RT-qPCR on a Mastercycler Ep-Realplex platform (Eppendorf, Hamburg, Germany), using Power SYBR Green reagents (Applied Biosystems, Foster City, CA, USA). Cycle conditions were 95 °C for 10 min, followed by 40 cycles of 95 °C for 15 s and 60 °C for 1 min. Quantified gene expression values were normalized against those of GUSB, U6, or GAPDH (Appendix A); when indicated, WPRE sequence [42] was evaluated as a subrogated indicator for vector presence or expression (Appendix A).

To detect and quantify the specific miRNA, TaqMan™ MicroRNA Assays (Thermo Fisher) were used according to the manufacturer’s instructions (Appendix A). The real-time PCR amplification was carried out in a QuantStudio real-time PCR system (Thermo Fisher Scientific) under the following conditions: 95 °C for 10 min, then 95 °C for 15 s, and 60 °C for 60 s for up to 40 cycles.

### 2.12. Western Blotting

Cells or exosomal populations were harvested in RIPA (radioimmunoprecipitation assay) lysis buffer, and an equal amount (20 μg) of lysates were separated by 10% SDS-PAGE. Proteins were transferred to PVDF membranes using the iBlot Dry Blotting System (Invitrogen). After incubation with primary and secondary antibodies, signals were developed using an ECL kit (GE Healthcare, Uppsala, Sweden). Appendix A includes the list of all primary and secondary antibodies used.

### 2.13. Myocardial Infarction and AAV Administration

MI was induced in mice by ligation of the left anterior descending coronary artery, as previously described [43]. Briefly, 8 to 10 weeks old mice were anesthetized with vaporized isoflurane, intubated using a 20 G intravenous catheter, mechanically ventilated, and placed on a heating pad to maintain body temperature. A left thoracotomy was performed via the fourth-fifth intercostal space, and the lungs were retracted to expose the heart. After opening the pericardium, the LAD coronary artery was ligated with a 7-0 non-absorbable ethylene suture approximately 2 mm below the edge of the left atrial appendage. After visual verification of anemia and akinesis of the apex and anterior-lateral wall to ensure coronary occlusion, the thorax was closed in layers. After extubation, mice were kept warm until fully recovered. AAV administration (1 × 10^11^ vg) was carried out by intramyocardial injection, 15 min after artery ligation, at two points of the peri-infarcted area by using a Hamilton syringe (Hamilton, 701 N, 10 µL). Electrophysiological evaluation of heart function was performed by echocardiography.

### 2.14. Echocardiographic Studies

Echocardiography was performed using a Vevo 3100 ultrasound system (Visualsonics, Manchester, UK), and measurements were optimized for small animals as previously described [44]. Echocardiography was performed at 2, 30, and 60 days after ligation of the left anterior descending artery. Left ventricular (LV) remodeling was quantified according to the guidelines and standards set by the American Society of Echocardiology, the Guide to Micro-Echocardiography Study using the Vevo3100, and the Vevo 3100^®^ Protocol-Based Measurements and Calculations guide.

### 2.15. Statistics

Assays were performed three times, and the data were expressed as mean ± SD. The normal distribution was assessed using the Shapiro-Wilk test and the D’agostino-Pearson test. Comparisons between/among groups were performed using the Student’s *t*-test and one-way ANOVA. Graphic representations were performed using Prism GraphPad 6.01 software, and statistical analyses were performed using Stata 12 for Windows software (College Station, TX, USA: StataCorp LP). Differences were considered statistically significant when the *p* value was (*** <0.001, ** <0.01, * <0.05).

## 3. Results

### 3.1. Comparative Proteomics Analysis of Adult Human Cardiac Progenitor Cell (CPC) Exosomal Compartment

Allogenic CPC has been demonstrated to promote significant improvements after MI and in heart failure in preclinical studies [3,4,5,6,45]. Most of these effects have been associated with secreted growth factors and cytokines, as well as EV, mainly exosomes [46]. To obtain a complete definition of the secreted proteome of human CPC [29] we have carried out a comparative analysis of their exosomal compartment. Independent CPC isolates, obtained as previously described [30], were cultured in conditions equivalent to those used in the associated clinical trial (CAREMI; EudraCT 2013-001358-81), using 3% O_2_, which was determined to be beneficial (critical in some contexts) for functional preservation and enhanced expansion rate of CPC [29,30,46]. We obtained a highly pure exosomal fraction as found by electron microscopy (Figure 2A,B), with a mean particle size of 170 nm (Figure 2C). The purified fraction showed expression, by western blot, of the CD63, CD9, HSP70, and CD81 typical exosome markers, although with some variations in the levels of expression among CPC1-CPC3 isolates (Figure 2D).

First, we confirmed that CPC exosome fractions (10^7^ or 10^8^ exosomes) had comparable angiogenic activity on HUVEC (Figure 3A). CPC secretome [29] also demonstrated a strong angiogenic capacity that seems to be a constant feature in most cardiac exosomal populations evaluated [22,47,48]. In addition, CPC exosomal preparations also demonstrated wound healing activity on human dermal fibroblasts (HDF) (unpublished data).

Proteomic analysis of the exosomal fraction was carried out by LC-MS/MS in three CPC isolates. Globally, after IPA analysis, we identified 1339 proteins common to the three isolates (Appendix A). The majority of the proteins identified were cytoplasmic (Appendix A), and only 508 of the common proteins identified in CPC exosomes were included in the microvesicles data base (Appendix A); enzymes were the most represented proteins (309 out of 1424), but the CPC exosomal compartment included more than 80 transcriptional regulators (Figure 3B). In addition, 9 cytokines and 7 growth factors, as well as 17 G protein-couple receptors (GPCR) and 20 transmembrane receptors (TMR), were also identified in the CPC exosomal fraction (Figure 3C). Ingenuity Pathway Analysis (IPA) demonstrated a high significant representation (*p* value < 1 × 10^−10^) of cardiovascular development functions (108 proteins), angiogenic functions (89 proteins), and different endothelial cell functions in the CPC exosomal compartment (Figure 3D). In addition, pathway analysis (Figure 3E) also revealed a prominent representation of the PDGF signaling pathway and other relevant pathways (including NGF, VEGF, HGF, and TGFβ), previously described as key for cardiovascular repair. These results are compatible with previous data on MSC exosomes [49], as CPC were previously described as an MSC-like cell population [31]. Comparative analysis of the CPC exosome proteome with the previously defined CPC secretome [29] confirmed the presence of several cytokines and growth factors, such as IL7 and EGF, that were found exclusively in exosomes (Appendix A).

### 3.2. Specific miRNA Repertoire of Human CPC Exosomal Compartment

Next, we aimed to define the human CPC miRNA repertoire (exo-miR) to infer global putative functional implications. CPC exosome analyses showed that the second most represented subfamily is miRNA (27.3%) (Figure 4A). From the 3 isolates, a total of 350 different miRNA (10 or more reads per million) were found in the CPC exosomal fraction (exo-miR), compared with the expression of 391 in the miRNA transcription profile of the whole CPC (Figure 4B,C); 292 miRNAs were common, whereas 60 and 96 were preferentially found in the exosomal compartment and the whole cell, respectively (Figure 4B,C; Appendix A).

Figure 4D illustrates the differential exo/cell expression profile for the CPC3 exo-miR repertoire. From the 30 top miRNAs, miR-2110 and miR-1911-3p were those miRNAs that showed the highest exo/cell ratio (23.5–25.5 × 10^3^-fold); they could be considered specific for the exosomal compartment. Also, miR-1268 was exclusively found in the exosome compartment (thus, no ratio could be represented). Then, to gain specificity in the association of exosomal miRNA with a putative cell function, CPC exo-miR was challenged with equivalent exosomal populations of human MSC and HDF exosomes as a distant reference. Both unsupervised hierarchical clustering (Figure 4E) and the Principal Component Analysis (PCA) (Figure 4F) neatly distinguished the three cell populations. The comparative analysis of the miRNA exosomal compartment for the three populations defined 40 miRNA that were differentially expressed in CPC vs. MSC and 48 in the comparison CPC vs. HDF (Figure 4G); then, 40 exosomal miRNA (exo-miR^SEL^) were commonly differentially expressed (up or downregulated) between CPC and both MSC/HDF (Figure 4G; Appendix A). Furthermore, among the common 40 CPC differentially expressed in exo-miR^SEL^, 12 miRNAs (30%) and 28 miRNAs (70%) were found up- and down-regulated in CPC exosomes (Figure 5A,B), respectively.

### 3.3. miR-935 in the Exosomal miRNA Repertoire of Murine Cardiac Progenitors

First, we tried to validate results obtained with human cardiac CPC with the well-characterized primary murine CPC cardiac population (B-CPC) [33,50]. This subpopulation was purified by cell cytometry from the hearts of the indicated mouse model (Appendix A) and compared with other heart populations (Appendix A). RT-qPCR analysis of miR-935 levels demonstrated that it was preferentially expressed by Bmi1^+^Sca1^+^ cells, whereas the CM-enriched population showed the lowest (<20-fold) levels (Figure 6B). This expression profile was demonstrated to be different from that of miR-300 (Appendix A), used as an internal control, and previously associated with murine cardiac immature progenitor regulation [51].

We have previously demonstrated that murine B-CPC are significantly resistant in vivo to AMI and other oxidative damage regimens, being protected in low-ROS perivascular (niche-like) domains [32]. Therefore, we evaluated the AMI impact (5d-post AMI) on miR-935 expression, finding a significant reduction in all populations analyzed (Figure 6B) and becoming almost undetectable in CM. The effect of AMI on miR-300 expression was again quite different (Appendix A), suggesting a substantially different role in B-CPC. In conclusion, miR-935 expression, although significantly overexpressed in murine Bmi1+Sca1+ cardiac progenitors, is severely affected by AMI in most cell lineages analyzed; this is not a general effect derived from AMI because reduced expression seems to be more pronounced on miR-935 in comparison with the impact on miR-300 expression (Appendix A).

For further characterization of the role of miR-935 in CPC, we used an immortalized cell line derived from murine B-CPC cells (Bmi1-CPC^IMM^). This line recapitulates most of the characterized parameters of Bmi1-Sca1+ cells and is labeled with YFP (Figure 1) [52]. In Bmi1-CPC^IMM^, miR-935 is overexpressed in the exosomal compartment (Figure 6C). Aiming to confirm the capacity of the Bmi1-CPC ^IMM^ exosomes to transfer miR-935, we first selected the HL-1 cell line, a murine CM-like cell line, that showed a level of intracellular miR-935 similar to that of Bmi1-CPC ^IMM^ cells (Appendix A). We confirmed that, compared with controls, co-culture of HL-1 cells with Bmi1-CPC^IMM^ cells promoted a significant increment in endogenous levels of miR-935 in HL-1 cells, sorted out after co-culture; this was not so evident by the addition of Bmi1-CPC ^IMM^-conditioned medium (Figure 6D). Finally, we evaluated miR-935 transfer to HL-1 cells under oxidative stress conditions (Figure 6D). Results indicated that miR-935 transfer was significantly enhanced both by Bmi1-CPC^IMM^-conditioned medium (3-fold) and co-culture with Bmi1-CPC^IMM^ (11-fold). Therefore, we concluded that miR-935 forms part of the mouse cardiac progenitor exo-miR and that its transfer to other cell types is significantly enhanced under oxidative stress conditions.

### 3.4. miR-935 Does Not Regulate Proliferation but It Is Involved in Antioxidative Damage Response

Gene Ontology (GO) analysis of the functional categories of target genes associated with the mouse (Figure 4E) and human (Appendix A) miR-935 found that general cellular processes such as proliferation and apoptotic signaling are included among the enriched functions, as well as cardiovascular development and angiogenesis. Few experimental studies on miR-935 function have been published currently, but a potential role as a tumor suppressor has been proposed in several cancer models [53].

We, therefore, first explored the potential role of miR-935 in regulating proliferation by transient transfection experiments in mCF and HL-1 cells using miR-935 mimic or antagomiR (INH) (Figure 6F,G). Results demonstrated that both miR-935 mimicking overexpression or inhibition did not significantly affect proliferation, so we concluded that miR-935 seems not to play a relevant role in proliferation in homeostasis regulation. GO bioinformatic analysis also indicated a potential implication in apoptotic signaling and cell death in response to oxidative stress. We, therefore, evaluated miR-935 in the context of cellular response to oxidative stress in human cells, as we previously showed that miR-935 transfer in the murine HL-1 cell line is significantly enhanced under oxidative stress (Figure 6D). To allow a correct interpretation of results, we first monitored the eventual modulation of miR-935 levels, comparatively, in the total cells and the exosomal compartments of CPC after H_2_O_2_ treatment (20–30 mM); these treatments did not provoke a reduction in miR-935 levels but a moderate increase (1.5–2 fold) in both compartments, although not statistically significant (Figure 7A); these results are in sharp contrast to those found in vivo after AMI (Figure 6B).

Because cardiac fibroblasts are a key population in heart damage response, we first evaluated the consequences of manipulating miR-935 levels in human cardiac fibroblasts. After miR-935 mimic or miR-935 antagomiR (INH) transfection, compared with controls (scramble), cells were exposed to oxidative stress (200 mM H_2_O_2_ during 48 h). Results clearly indicated that whereas miR-935 mimic overexpression did not modify the response to oxidative stress damage, compared with controls (Figure 7B; Appendix A), transfection of antagomir (INH) induced a significant increment, both in apoptotic (Annx^+^ PI^−^) and necrotic (Annx^+^ PI^+^) populations (Figure 7B); after transfection of INH, the increment in percentage of Annx+ and IP+ cardiac fibroblasts (Appendix A) showed similar levels (67 and 70%, respectively). Comparable results were obtained in HL-1 cells (Figure 7C); INH transfection promoted apoptosis (Annx^+^ PI^−^) but again, transfection of the miR-935 mimic did not provoke any significant effect on survival upon exposure to H_2_O_2_ (100–200 mM H_2_O_2_, during 48 h). Next, using a protein array designed for detection of proteins involved in apoptosis, we compared HL-1 cells subjected to oxidative stress, pre-transfected with INH, and transfected with the scramble construct. Results found that 70% of proteins included in the array were upregulated in HL-1 cells pre-transfected with INH; only BCL-W was found to be moderately downregulated (Figure 7D).

Among the more upregulated proteins, we found IGFBP4 (9.5-fold), cIAP2 (6.5-fold), CASP8 (6.5-fold), and DR6 (5-fold) (Figure 7D). Most of these up-regulated proteins are predominantly involved in cell death mechanisms such as apoptosis (DR6, CASP8) or senescence (IGFBP4). On the contrary, cIAP2/HIAP1 regulates not only caspases and apoptosis but also inflammatory signaling.

These results, therefore, demonstrated that under moderate oxidative damage in several cell lines, miR-935 levels were not significantly modulated, but reduction of miR-935 levels was associated with the induction of oxidative-stress-associated apoptosis. This suggests that miR-935 could be participating in the pathway(s) involved in antioxidant stress defense.

### 3.5. Evaluation of Putative miR-935 Targets in the Oxidative Damage Context

Several miR-935 targets, evaluated in other cellular models or predicted in silico, were defined (Appendix A). We selected NOTCH1 and SOX7 (validated targets) for experimental evaluation and ARNT2 (Aryl Hydrocarbon Receptor Nuclear Translocator 2; HIF-2α) and CASP10 (Caspase 10) as in silico predicted targets; ARNT2-miR-935 was also predicted by correlation of main pair (miRNA:mRNA) analysis, comparing the potential of overexpressed miRNA in CPC to putatively targeting transcriptional factors (Appendix A).

We first analyzed the impact on the expression level of these four putative miR-935 targets in HDF cells after co-culture with purified CPC exosomes. In comparison with their basal expression levels, *ARNT2*, *NOTCH1* and *CASP10* did not show a significant reduction, and *SOX7* expression seemed not to be affected (Figure 8A). After miR-935 mimic transfection in HDF cells, in homeostasis, two putative targets clearly reduced their expression (*ARNT2* and *SOX7*); *CASP10* showed a similar response, but it was not statistically significant. On the contrary, *NOTCH1* did not show modulation by miR-935 mimic overexpression. In addition, miR-935 INH transfection in HDF cells only promoted a moderate trend to up-regulation for the four target genes (1.5–3.5 fold) but was not statistically significant (Appendix A); we confirmed that these results were not exclusively related to differences in basal expression levels of each putative target in HDF [30].

In order to evaluate the relevance of the miR-935 target in the oxidative stress context in HDF, we first analyzed the direct modulation of the expression of the four putative miR-935 targets in response to oxidative stress (100 mM H_2_O_2_) to assist in the results’ interpretation. Results demonstrated that all of them were significantly up-regulated upon oxidative stress induction (Figure 8B) in comparison with basal conditions. Analysis of the response of these putative miR-935 target genes after oxidative damage in HDF-mimic transfected cells showed significant downregulation of *NOTCH1* and *ARNT2*; *SOX7* demonstrated only a trend for reduced expression (Figure 8C). In addition, HDF subjected to oxidative damage after transfection with miR-935 INH only showed a moderate trend for *NOTCH1* upregulation (20–35% upregulation) but was non-statistically significant. These results could be related to the low level of up-regulation of *NOTCH1* and *ARNT2* provoked by oxidative damage in HDF (Figure 8B).

All putative targets evaluated were regulated by miR-935 in human fibroblasts, suggesting that its overexpression (or concerted action with additional miRNA) could participate in the prevention of apoptosis/necrosis observed in HL-1 and human cardiac fibroblasts upon inhibition of miR-935 in the context of oxidative damage (Figure 7B,C).

### 3.6. Evaluation of Cardioprotection Activity of miR-935 in a Mouse Model of Acute Myocardial Infarct

Based on the previous data obtained with the human model of CPC, the confirmed high levels of expression in murine cardiac progenitors (B-CPC cells), and the clear reduction in miR-935 levels after AMI, especially in CM (Figure 6B), we aimed to evaluate a putative in vivo protective role of miR-935 upon overexpression in infarcted animals using cardiotropic AAV9 vectors.

We hypothesized that partially counteracting the severe reduction of miR-935 levels as a consequence of AMI could positively impact cardiac recovery. The vector AAV-TnT-miR935 (AAV-935) and a control vector (AAV-control) that expresses a scrambled miRNA sequence were used (Figure 8D). In the first set of experiments, results clearly indicated that intramyocardial administration of AAV-935 was efficient, as estimated by RT-qPCR detection of AAV vectors in purified CM (Appendix A). AAV-935 and AAV-control viral vectors were injected in the peri-infarct area of mice immediately after the induction of the infarct, and animals’ cardiac function was analyzed 2 and 60 days after AMI (Figure 8E). miR-935 overexpression was also confirmed by detecting higher miR-935 expression levels (>2-fold) in the infarcted area of the hearts in comparison with the remote areas (Appendix A).

A significant functional improvement was not found after miR-935 treatment, despite the fact that a slight positive trend in cardiac EF was observed after 2 months of miR-935-AAV treatment in comparison with the control group (Figure 8F). All these results suggest that the overexpressed miR-935 levels in vivo were not sufficient for a therapeutic effect in the context of severe cardiac damage (AMI). Probably, the combination of miR-935 with other cardioprotective exosomal miRNAs could act in concert to counteract oxidative stress-related apoptosis induced after MI.

## 4. Discussion

Human adult cardiac progenitors (CPC/CDC) promote functional improvements upon transplantation in large infarcted animals [3,5,6]. Most of these effects have been associated with secreted growth factors and cytokines, as well as cargoes of EV, mainly exosomes [24]. The relevance and differential potential of the exosomal compartment are remarked by the fact that up to 10% of the proteins identified by LC-MS/MS proteomics in the CPC exosomal compartment were not found in the CPC secretome, independently of their level of expression [29]. IPA confirmed that CPC-exosomes showed a highly significant angiogenic profile and plausible involvement in cardiovascular repair and inflammation regulation through relevant signaling pathways. These results are also compatible with previous data on MSC exosomes [49], as CPC were previously described as an MSC-like population [31].

In addition, the evaluation of the CPC miRNA repertoire allowed us to infer some global functional implications. A total of 350 different exosomal miRNAs (exo-miR) were found, and bioinformatic analysis defined more than 2200 genes as potential targets of exo-miR, including Actin Alpha Cardiac Muscle 1 (ACTC1) and Calcium Voltage-Gated Channel Subunit Alpha1 C (CACNA1C), regulated with high probability by 5 and 7 exo-miR, respectively (unpublished data). In addition, exo-miR^SEL^ has the potential to regulate a relevant number of transcription factors (Appendix A). Finally, many exo-miR^SEL^ have been associated with positive and negative implications, both in homeostasis regulation and different cancers (Appendix A).

Taking together both proteomic and mRNA signatures, CPC-exosomes showed a highly significant angiogenic profile and plausible, direct or indirect, involvement in cardiovascular physiology regulation and repair, contributing to heart repair after damage. A first conclusion noted a high variable regulation potential but with several miRNA^SEL^ elements displaying antagonistic roles, most probably depending on the specific futures of each cell type and even conditioned by external signals (e.g., niche integration).

Within the exo-miR population, 58 were preferentially represented in the CPC exosomal compartment. MiR-2110 and miR-1911-3p were those that showed the highest exo/cell ratio, being considered almost specific for the exosomal compartment (Appendix A). Both have been associated with several forms of human cancer [53,54,55]. Aiming to define the most specific/p B-CPCIMM cell-tial signature of human CPC exosomes, we compared exo-miR human CPC with human bone marrow MSC- and HDF-exosome profiles. This analysis defined 40 exo-miR that were differentially expressed (commonly downregulated) between CPC and MSC/HDF and 12 miRNA (Figure 5A,B; Appendix A) that were overrepresented in CPC exo-miR.

Comparison of the levels of expression of exo-miR^SEL^ in exosomes with the whole cell levels (exo/cells ratio) indicated that miR-1268 is almost exclusively found in the exosomal compartment, but no specific function has yet been ascribed to miR-1268, although some weak association has been reported in prostate cancer [55]. Interestingly, miR-625-3p, miR-146a-3p, and miR-135b-5p also showed preferential exosomal representation (exo/cell ratios of 7.7, 3.4, and 2.0, respectively). In previous studies, miR-146a has also been identified as one of the enriched miRNAs in CDC exosomes, and it has been demonstrated to exert a beneficial effect upon transplantation in infarcted animals [56]. In our analysis, miR-146a-3p is clearly overexpressed in exo-miR^SEL^ but with a low-to-medium absolute level of expression. Concerning miR-625-3p, serum levels have only been linked with cholesterol levels in the metabolic syndrome and the cardiovascular field [57]. In addition, miR-625-3p has been proposed as a predictive parameter for successful lymphohematopoietic reconstitution after allogenic bone marrow transplantation, both in mice and humans; in this basic and clinical procedures, miR-625-3p expression in exosomes was not evaluated [58]; more recently, it has also been associated with human cancer [59]. Finally, miR-135b-5p has been implicated in the promotion of gastric and pancreatic cancer [60].

The mechanism(s) by which miRNAs get sorted into exosomes or retained in cells are not fully understood. Recently, it has been demonstrated that miRNAs possess sorting sequences (EXOmotifs) that facilitate their secretion loading in exosomes in a cell-dependent way. The global conclusions stated that miRNAs that display a higher CG content have a higher probability of being enriched in exosomes [61]. The analysis of the highly preferential exo-miR^SEL^ population showed that the above-mentioned code fit well with the miRNA with the highest differential representation in CPC exosomes. The analysis of miRNAs that are highly overrepresented in CPC relative to MSC/HDF, independently of their preferential sorting to exosomes (Appendix A), confirmed that miRNA-1268, mostly specific for CPC exosomes, showed the highest CG content (83.33%) among all exo-miR^SEL^ populations. In conclusion, in CPC, the high CG content combined with the presence of some EXOmotifs are good criteria to predict miRNAs that show the highest exosomal sorting potential. Additional factors should cooperate for miRNA sorting to exosomes of medium-level expression miRNA [61].

Then, as a first attempt to characterize the functional role of some exo-miR^SEL^, we focused our study on miR-935, which showed the highest differential expression ratio between CPC and MSC/HDF, both represented in the whole cell and the exosomal compartment (exo-miR^SEL^). Altered levels have been again associated with several human cancer types, mostly relating miR-935 overexpression with malignant stages [62]. In osteosarcoma, however, miR-935 inhibits proliferation and invasion [63]. Therefore, it seems that miR-935 function could vary substantially depending on the cellular and even physiological/metabolic contexts. Regarding the latter, it is relevant that, in human colorectal cancer, miR-935 is significantly sorted to exosomes, but in a cellular mutant KRAS-dependent manner, therefore suggesting a new role for KRAS in the sorting of miRNA to exosomes [64]. This adds another layer of complexity to the functional validation of miR-935′s role that could vary substantially in homeostasis, under damage, or under specific pathological conditions, such as colorectal cancer. Only very recently has miR-935 been confirmed as a relevant determinant in MSC-derived exosomes that favor proliferation and differentiation of osteoblasts upon co-culture [65].

In our cell system, however, we showed that manipulation of miR-935 levels did not significantly affect proliferation or survival in different cell lineages. Gene Ontology bioinformatic analysis also suggested a potential implication in regulating cell death in response to oxidative stress. Therefore, we explored the potential role of miR-935 under oxidative stress conditions, finding a clear implication in the responses of several cell lineages. These results suggested that miR-935 could be a relevant actor for an anti-apoptotic response associated with moderate oxidative stress. Protein array analysis on HL-1 cells confirmed that transfection of the miR-935 inhibitor (INH) coupled to moderate oxidative stress provoked a significant upregulation of an important number of apoptosis mediators. It is worth considering that miR-935 levels in CPC are high and are not reduced by moderate oxidative stress, neither in whole cells nor exosomes; this could be an interesting attribute for participating in such an anti-oxidative stress pathway. A few additional examples of exosomal miRNA and circular RNAs, both cardiac or from heterologous cells, have been demonstrated to participate in regulating oxidative stress-related apoptosis (reviewed in [66]). Concerning those miRNAs specifically expressed by progenitor cells, like miR-21, miR-21-5p, miR-126-3p, miR-146a-5p, and miR-210, they have been shown to inhibit apoptosis, necrosis, or oxidative stress, being upregulated in exosomes after oxidative stress induction. In comparison with miR-935, only miR-146a-5p demonstrated significant high exosomal levels; miR-146a-5p is the main player involved in protection against doxorrubicin-induced cell death [56]. MiR-210 is majoritarian cellular, and miR-21-5p is poorly represented in CPC. Therefore, miR-935 and miR-146a-5p could act in concert with other global exosomal secreted miRNAs, such as miR-126, to counteract oxidative stress-related damage.

miR-935 selected targets (both confirmed and bioinformatically predicted targets) were evaluated in HDF under oxidative stress conditions (Figure 8C). Only *NOTCH1* showed a trend to be upregulated upon miR-935 inhibitor (INH) transfection. Transfection with miRNA-935 mimic (25 nM) significantly reduced the expression of *NOTCH1* and *ARNT2*, with a clear similar trend for *SOX7* (Figure 8C). The evaluation of the selected miR-935 targets, both in homeostasis (Appendix A) and under oxidative stress conditions (Figure 8C), demonstrated that miR-935 mimic and inhibitor (miR-935-INH) transient overexpression provoked the expected modulations in agreement with previous results, with the exception of *NOTCH1*, whose expression was not reduced after miR-935 mimic transfection in response to oxidative stress.

In mouse myocardium, *Notch1* has a critical function during development, but in adult hearts, Notch1 signaling seems to be only activated in response to several forms of damage, favoring angiogenesis and demonstrating cardioprotective effects [67]. Therefore, in homeostasis, it could be hypothesized that CPC-derived exosomes (rich in miR-935) could be contributing, in coordination with intracellular levels of miR-935, to controlling the basal levels of these key integrative stress/injury signal pathways. CPC exosomes are also rich in miR-146-3p, which also targets Notch1 [68], so the miR-935 action could be reinforced. In addition, the exosomal miR-935 could also impact several cardiac cell lineages and some heart infiltrating populations for the orchestration of damage responses (TLR4-miR-935 pathway); it seems to be involved in modulating the activation of infiltrating monocytes and their M2-like macrophage differentiation, as recently reported [69].

Aryl Hydrocarbon Receptor Nuclear Translocator 2 (ARNT2; HIF2α), regulates the translocation of the ligand-bound aryl hydrocarbon receptor (AhR) complex to the nucleus, modulating sensitivity to apoptosis [70,71,72]. Whether ARNT2 should have a similar role in CPC, miR-935 could also be modulating apoptosis sensitivity through ARNT2 downregulation; in glioblastoma, ARNT2 plays a central role in regulating stem-like cell tumorigenic properties in vivo [73].

Interestingly, a link between NOTCH1 and ARNT2 has been proposed that depends on AhR heterodimerization with ARNT or ARNT2 for signaling (through IL22), both in homeostasis or in acute inflammation and regeneration. NOTCH1 favors expression of ligands (ARNT/ARNT2) for the aryl hydrocarbon receptor (AhR) (reviewed in [74]) Therefore, miR-935 has the potential to regulate this pathway that involves both NOTCH1 and ARNT2; no relevant functional information is available for ARNT2 in the adult cardiovascular system, both in mice and humans.

Due to the profiles obtained in HDF (pre-transfected with mimic or INH), in homeostasis, and after the induction of oxidative stress, only the expression profile of *CASP10/Casp8* can be directly associated with the phenotype described. Upon oxidative stress induction, *CASP10* expression is highly augmented in HDF (more than 2 × 10^4^ fold), so the level of inhibition provoked by mimic overexpression, although not statistically significant, could be considered a relevant response with regard to the modulation of apoptosis/necrosis coupled to oxidative stress.

Aiming to confirm the proposed role for miR-935 in B-CPC [32] we found that miR-935 is preferentially overexpressed in the B-CPC in comparison with other heart populations, being severely downregulated in all populations analyzed post-AMI. This implies that early after AMI, miR-935 targets could be significantly upregulated, eventually contributing to the severity or extension of the damage. The comparison of the results obtained in B-CPC (in vivo, in response to AMI) and CPC (moderate oxidative stress) indicated that there is no clear parallelism, both in the modulation of miR-935 levels and target genes. In the complex scenario post-AMI, although no impact on B-CPC numbers was found [32], there is no clear inverse correlation between miR-935 level and the analyzed targets in the B-CPC population. The simplest explanation could be that AMI could constitute much more severe and complex damage compared with the in vitro evaluation of oxidative damage. Additional explanations could be related to differences in miR-935 biogenesis or to a significant increment in exosome secretion coupled to AMI, as previously reported for other miRNAs [75]. In both scenarios, the B-CPC population could appear to be miR-935 depleted. Because we have found that miR-935 does not have a direct involvement in proliferation in several target cell populations (including CPC), it could be hypothesized that downregulation of miR-935 could form part of in vivo signals involved in initiating B-CPC commitment/differentiation, as demonstrated previously in B-CPC upon AMI induction [32].

Interestingly, Zhang et al. reported, in a rat osteoporosis model, a significant decline in miR-935 levels associated with ovariotomy; reconstitution of miR-935 levels reduced the severity of osteoporosis [65]. Based on these recent and encouraging data and the information obtained in human and murine cardiac progenitors, we aimed to evaluate a putative in vivo cardioprotection role of miR-935 upon overexpression with AAV9-TnT vectors in infarcted animals, as previously reported [41]. We hypothesized that partial counteracting the severe reduction of miR-935 levels as a consequence of AMI could impact cardiac damage and eventually some recovery. Results, however, suggested that we have not been able to induce relevant therapeutic miR-935 levels in vivo in the context of AMI; only a trend of improvement in cardiac function was found. Accordingly, evaluation of many of the plausible miR-935 targets in heart samples did not get modulated as expected (unpublished data).

Our study, therefore, possesses some limitations, as we could not confirm the expected in vivo cardioprotective role of miR-935 when injecting the AAV-935 vectors into the ischemic hearts. Probably, the combination of miR-935 with other described cardioprotective miRNAs such as miR145-5p [23,57], and/or others (miR-210, miR-21-3p, and miR-21-5p) could act in concert to counteract oxidative stress-related apoptosis induced after MI. Further dedicated research is required to demonstrate their in vivo role in the ischemic cardiac context. Another issue to be mentioned is the fact that in order to obtain a sufficient amount of exosomes for analysis, CPC need to be substantially expanded, which could eventually affect their exosomal profile. The confirmation of miR-935 expression in the murine CPC compartment supports the main conclusions with human cells.

## 5. Conclusions

A preferential exosomal miRNA repertoire (exo-miR^SEL^) has been defined in human CPC. MiR-935 has been identified as one of the most overrepresented miRNAs in exo-miR^SEL^, defined by comparison with reference cell lineages; this profile has also been confirmed in murine cardiac progenitors (B-CPC). In vitro functional evaluation strongly suggests that miR-935 plays a relevant role in the oxidative stress defense response, and its inhibition, combined with oxidative stress regimens, promotes apoptosis/necrosis in several cell lineages, although with significant differences among them.

## Figures and Tables

**Figure 1 cells-12-02300-f001:**
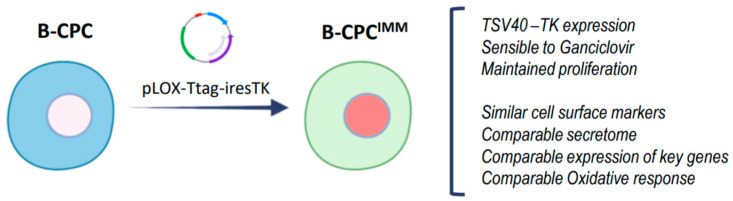
The main characteristics of the reversible SV40-immortalized B-CPC^IMM^ cell line, compared with their parental cells (B-CPC), were that pLOX-Ttag-iresTKm (Addgene, Watertown, MA, USA) was used, and B-CPC^IMM^ cells express YFP.

**Figure 2 cells-12-02300-f002:**
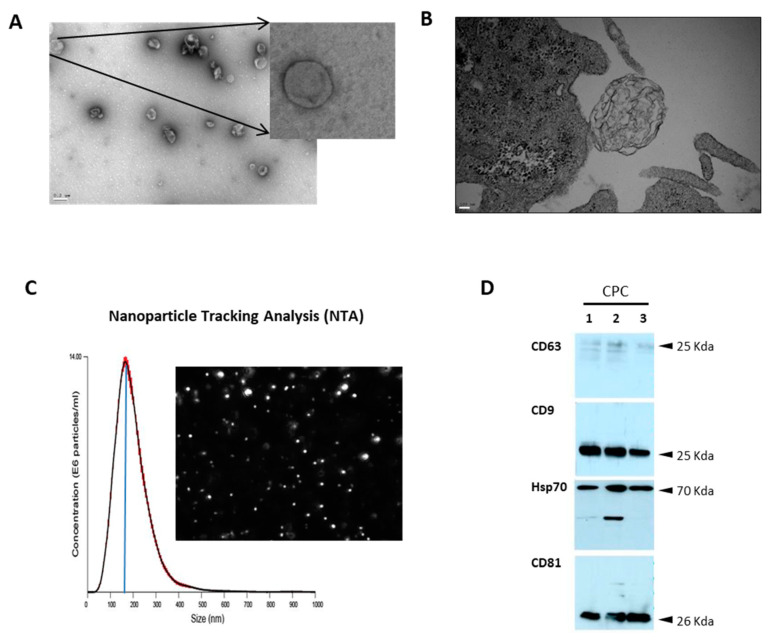
Characterization of CPC exosomal purified fractions (**A**,**B**) Transmission Electron Microscopy (EM) of CPC exosomes, n = 3; (**A**) Scale bar, 0.2 µM; inset, ×-10; (**B**) Scale bar, 100 nm (**C**) Nanosight tracking analysis and (**D**) western blot characterization of exosome markers in 3 different CPC isolates (CPC1-CPC3).

**Figure 3 cells-12-02300-f003:**
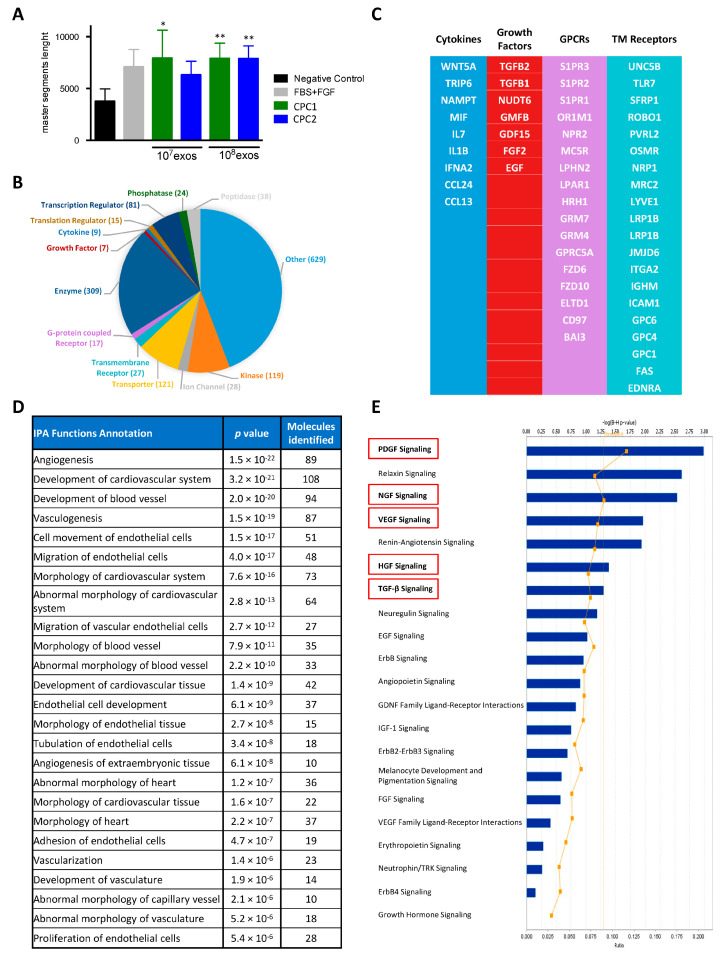
Proteomic analysis of the CPC exosomal compartment (**A**) Angiogenic activity of CPC (isolates CPC1 & CPC2) exosome preparations (10^7^ and 10^8^ exosomal particles), was evaluated in HUVEC. Negative (serum-free medium 199) and positive (FBS + FGF in medium 199) controls were included. After 6 h of incubation, tube formation was analyzed using ImageJ software. Data are expressed as the mean ± SD (n = 3); *p* values were (**) <0.01; (*) <0.05. (**B**) After whole label-free proteomic analysis of three independent CPC isolates (CPC1-CPC3), the distribution of the CPC exosome proteome (1339 proteins) was organized by functional activity. (**C**) The most relevant cytokines and growth factors (left) and GPCRs and transmembrane receptors (right) are indicated. (**D**,**E**) Ingenuity Pathway Analysis (IPA) of the CPC exosomal compartment demonstrated high significant representation (*p* value < 10^−15^) of several cellular functions (**D**) as well as prominent signaling pathways (**E**).

**Figure 4 cells-12-02300-f004:**
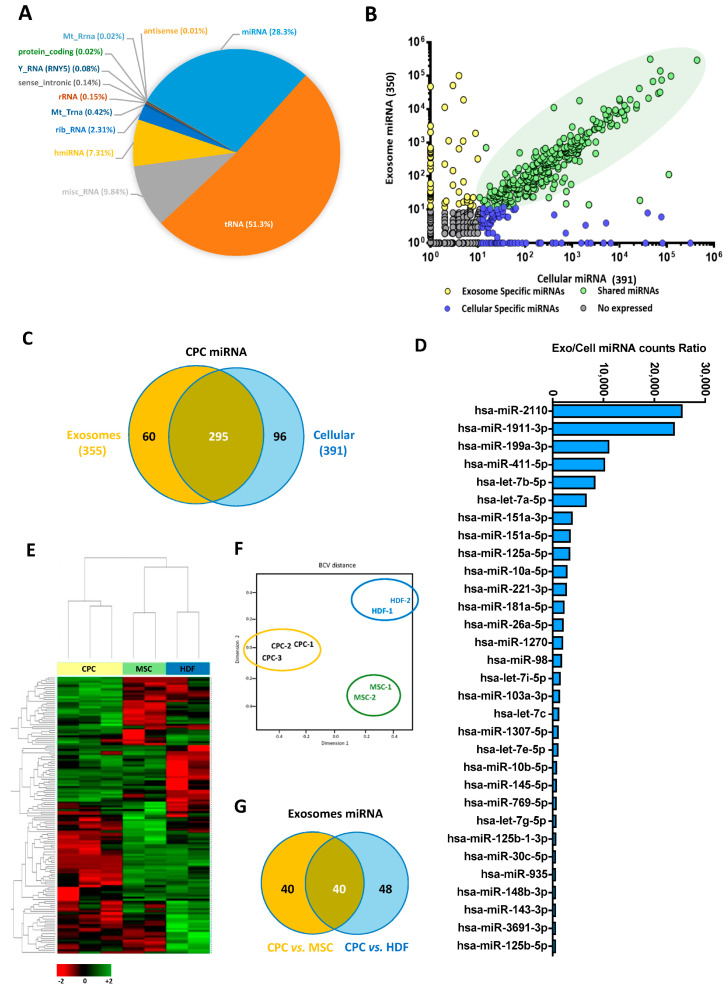
Comparative analysis of the exosomal miRNA repertoire of CPC (exo-miR) (**A**–**C**) RNA-seq analysis of whole cell CPC compared with the exosomal compartment was carried out. The distribution of miRNA in the context of total RNA in exosomes is indicated (**A**), and the comparison of the miRNA exosomal repertoire (exo-miR) with whole cell representation is represented (**B**); there are miRNA common to both compartments (green dots), mainly associated with the whole cell (blue dots), and miRNA preferentially present in exosomes (yellow dots); it is considered that a specific miRNA is differentially expressed when there is a difference > 10 counts/between samples. (**C**) Representation of the global distribution of the CPC miRNA comparing those more specific for exosomes (58) with the whole cell (99). (**D**) List of the main CPC miRNA, preferentially associated with the exosome compartment, organized by exo/cell expression ratio. RNAseq analysis of exosomal CPC miRNA compared with MSC and HDF Analysis of three CPC isolates (CPC1-CPC3) was compared with two MSC (MSC1, MSC2) and two HDF isolates (HDF-1, HDF2). (**E**,**F**) Normalized heat map analysis of a total of 481 miRNAs (**E**) and clustering analysis (**F**) confirmed that CPC, MSC, and HDF cell lineages are quite distant and represent significantly differentiated clusters. (**G**) Venn diagram representation of differentially expressed miRNA: the specific DEG CPC vs. MSC (orange), DEG CPC vs. HDF (blue), and common differentially expressed miRNA (40) are represented in brown.

**Figure 5 cells-12-02300-f005:**
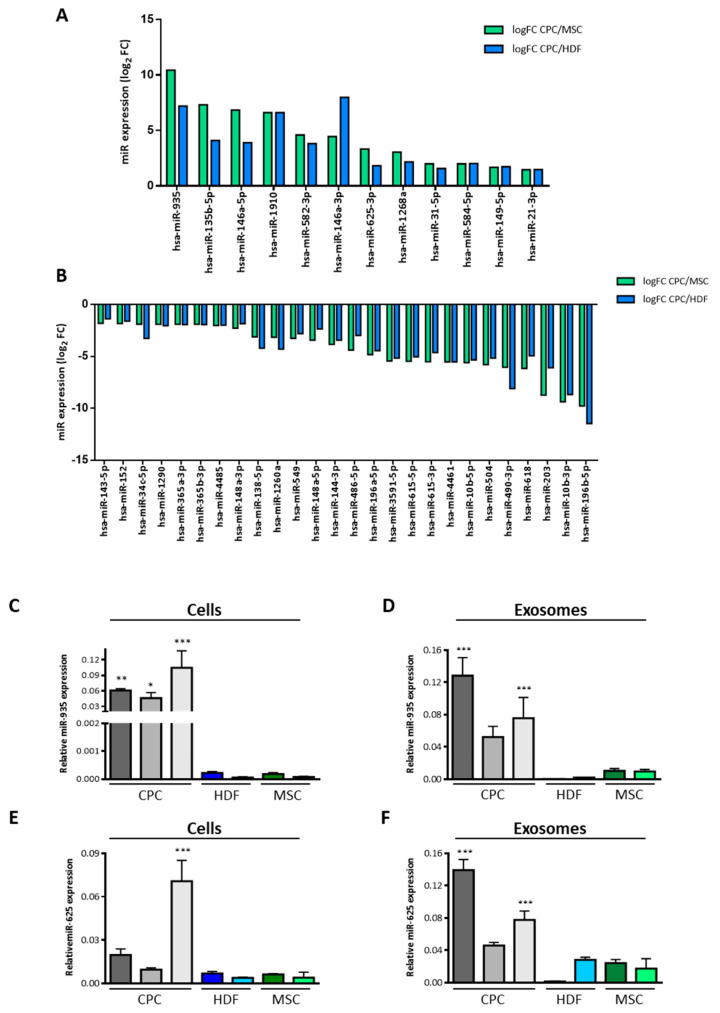
Identification and validation of the exosomal miRNA differentially expressed in CPC relative to the reference cells (**A**,**B**) Plot (log2 FC) of the main up-regulated (**A**) or down-regulated (**B**) miRNAs in human CPC vs. human MSC and HDF. (**C**–**F**) Comparative RT-qPCR analysis for miR-935 (**C**,**D**) and miR-625-3p (**E**,**F**) expression both in whole cells (**C**,**E**) or in exosomal preparations (**D**,**F**) of three CPC isolates (CPC-1–CPC-3), two MSC (MSC1, MSC2), and two HDF isolates (HDF-1, HDF2). Relative expression (normalized to GAPDH in whole cell analyses and in exosomal preparations) is expressed as mean ± SD (n = 3); *p* values were (***) <0.001; (**) <0.01; (*) <0.05.

**Figure 6 cells-12-02300-f006:**
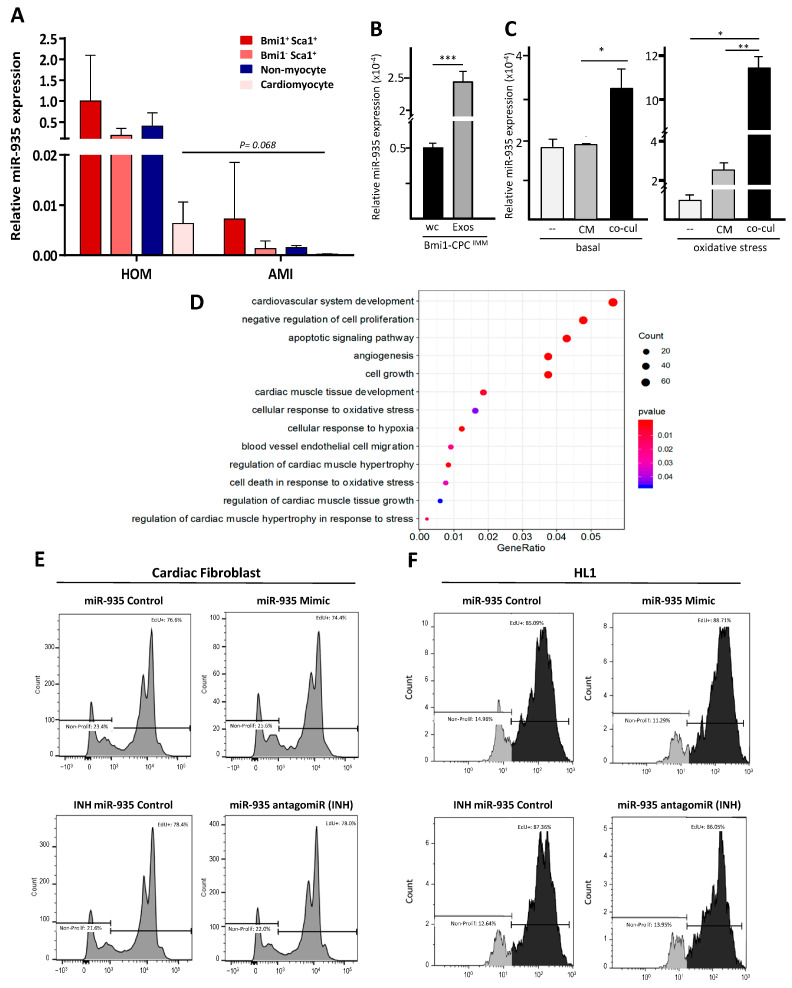
Evaluation of miR-935 expression in murine CPC and its involvement in proliferation regulation (**A**) RT-qPCR analysis of miR-935 expression in the indicated cardiac subpopulations, in homeostasis (HOM; left) or 5 days post-AMI (AMI; right). (**B**) RT-qPCR miR-935 expression analysis in the Bmi1-CPC^IMM^ line in the whole cell (WC) compared with the whole exosomal compartment (Exos). (**C**) Evaluation of miR-935 transfer in HL-1 cells after co-culture with Bmi1-CPC^IMM^ cells (co-cul) and compared with the supplementation of Bmi1-CPC^IMM^ conditioned medium (CoM), both in homeostasis (basal; left) or under oxidative stress (H_2_O_2_,100 µM; 48 h) condition (right); (--) indicates the negative controls. All relative data, normalized to GAPDH, are expressed as the mean ± SD (n = 3); *p* values were (***) <0.001; (**) <0.01; (*) <0.05. (**D**) Gene Ontology (GO) analysis of the functional categories of target genes associated with the murine miR-935; only categories with an enrichment *p* value < 0.05 were selected. (**E**,**F**) Analysis of the proliferation rate in mouse cardiac fibroblasts (**E**) and HL-1 (**F**) by Edu or BrdU incorporation, respectively; cells were previously transfected with miR-935 or the antigomiR (INH) and compared with cells transfected with the corresponding controls (scramble). Histograms indicate the percentage of proliferating cells in each condition.

**Figure 7 cells-12-02300-f007:**
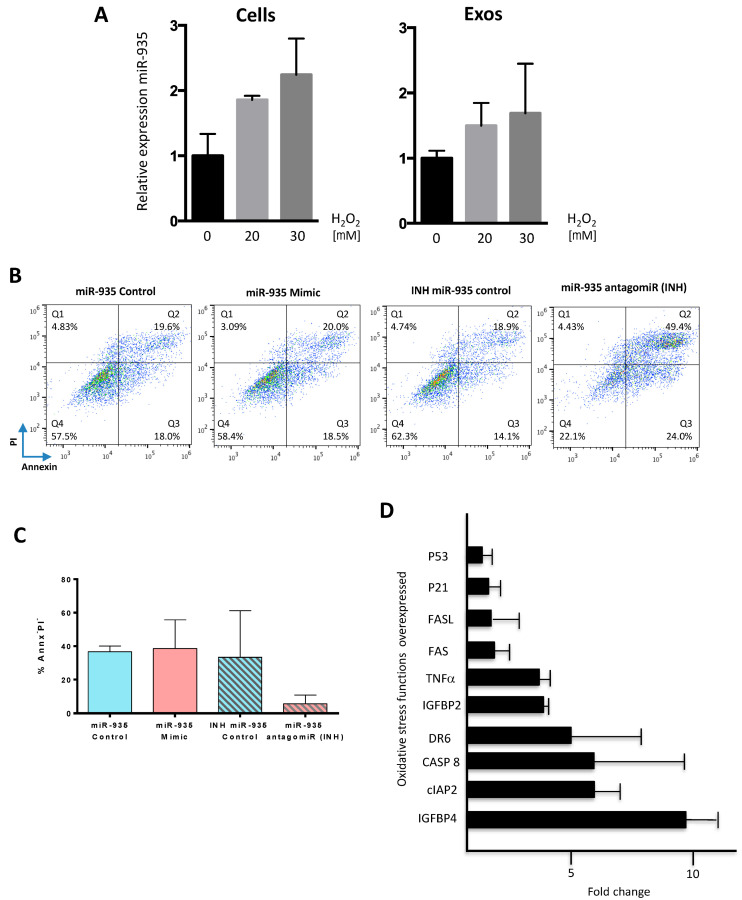
miR-935 and antagomiR responses under oxidative stress conditions (**A**) RT-qPCR evaluation of miR-935 levels, both in whole CPC (cells) and their derived exosomal compartment (exos), in response to different concentrations (0–30 mM) of H_2_O_2_, for 24 h. (**B**) Cardiac fibroblasts, prior transfected (25 nM each) with miR-935 mimic, the antagomiR (INH), or the corresponding controls (scramble), were cultured for 24 h, and then cells were exposed to H_2_O_2_ (100 mM) for an additional 24 h prior to analysis by cell cytometry using propidium iodide (PI) and annexin to define the apoptotic (Annexin^+^ PI^−^; Q3) and necrotic (Annexin^+^ PI^+^; Q2) populations; results are presented as dot blots. (**C**) HL-1 cells, first transfected with miR-935 mimic, the antagomiR (INH) or the corresponding controls (scramble) (25 nM each), were maintained in culture for 24 or 48 h, to be next exposed to H_2_O_2_ (100 mM) for an additional 24 or 48 h, prior to cell cytometry analysis, to evaluate the percentage of viable cells (Annx^-^PI^-^) using propidium iodide and annexin; data are expressed as mean ± SD (n = 3). (**D**). Protein array analysis of HL-1 cells pre-transfected with antagomiR (INH) and subjected to oxidative stress (100 mM) during 48 h was compared with that of control HL-1 cells (transfected with the scramble construct and also subjected to oxidative damage); the figure shows the most upregulated proteins (fold change).

**Figure 8 cells-12-02300-f008:**
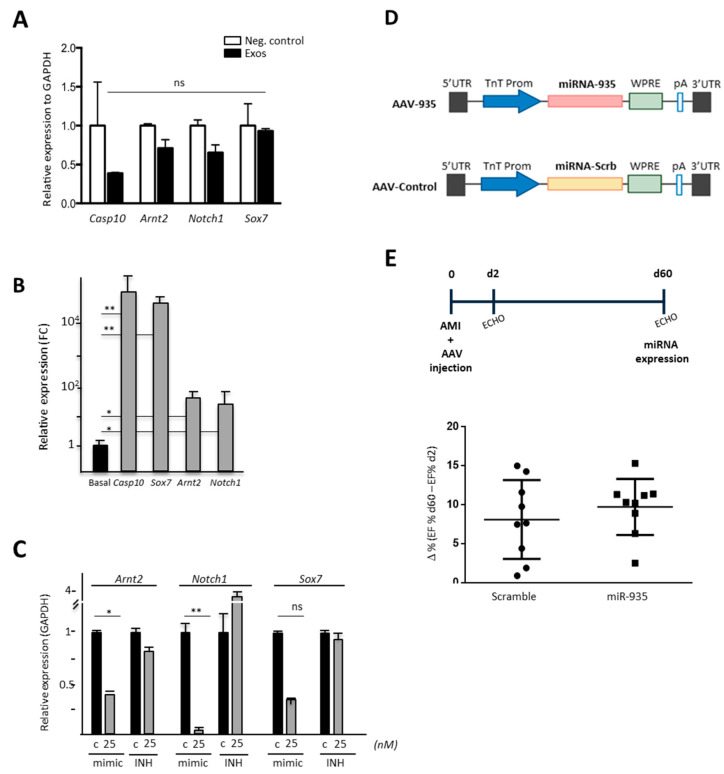
In vitro evaluation of potential miR-935 targets and in vivo cardioprotection capacity (**A**) RT-qPCR analysis of the indicated putative target genes in HDF after co-culture (48 h) with purified CPC exosomes (10^9^ exosomal particles). (**B**) RT-qPCR analysis of the indicated putative miR-935 target genes in HDF, subjected to oxidative stress (H_2_O_2_; 100 µM), during 24 h; *p* values were (**) <0.01; (*) <0.05; ns, no statistically significant. (**C**) HDF were transfected with miR-935 (mimic) or the antagomiR (INH) (25 nM each); the corresponding negative controls (scramble; c) were also transfected. The cultures were maintained for 24 h and then subjected to oxidative stress treatment (H_2_O_2_; 100 mM) for 24 h. RT-qPCR of the indicated putative target genes was carried out. (**D**,**E**) In vivo cardioprotection evaluation of miR-935 using AAV9-TnT vectors (**D**) Schematic representation of AVV vectors (**E**) In vivo experimental plan scheme (up). Immediately after AMI, intramyocardial administration of AAV-935 and AAV-control viral vectors (n = 9/group; 1.0 × 10^11^ vg/mouse) was carried out. Cardiac function was followed up in both groups of animals by echocardiography (ECO) at d2 and d60 post-MI; the percentage of differential Ejection Fraction (ΔEF) (EF% day 60–EF% day 2) is represented for the AAV-935 inoculated and control animal groups (down).

## Data Availability

All proteomics data derived from this study were deposited in Peptide Atlas (http://www.peptideatlas.org/repository/, accessed on 1 August 2023) and are accessible through the accession code PASS00827. All transcriptomic (miRNA) data are deposited in the Gene Expression Omnibus (GEO) database (http://www.ncbi.nlm.nih.gov/geo/query/, accessed on 1 August 2023) and are accessible through the accession number GSE84070. The rest of the analytical datasets generated during and/or analyzed during the current study are available from the corresponding author on reasonable request. The materials behind the conclusions of this study are available from the corresponding author upon reasonable request.

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
