# Peer review of "Cardiac Progenitor Cell Exosomal miR-935 Protects against Oxidative Stress"

_cells, 2023, doi:10.3390/cells12182300_

Round 1
Reviewer 1 Report
Aguilar et al reported MIR-935 is as one of most overrepresented mIRNA in exo- 834 miRSEL. Additionally, they showed that inhibition of miR-935 combined with oxidative stress treatment increases apoptosis/necrosis. Manuscript is well written. I have some suggestions.
1. Quality and resolution of all figures are low. Please fix them.
2. It would be great if the authors mention limitations of this research in discussion section.
3. I suggest that the authors write human CPC instead of human cardiac CPC in the text.
4. Please provide a reference for used concentrations of H2O2 (100mM-20mM) in material section.
5. Grammar needs to be connected throughout the paper.
6. The authors did not report any data on Wound Healing Assay. I suggest that the authors remove this section from materials& methods.
Minor editing of English language required.
Author Response
Please find the reply as attached

Reviewer 2 Report
Comments:
The scientific work is good and has lots of research outputs. However, the authors need to improve their image quality. The current version is not clearly visible and couldn’t understand the outcome of the study. So, all the image's visibility should be improved to get the acceptance. Also, there are some font size and spelling error appears in the manuscript. Along with this, address the below comments as well.
1. Mention the conditioned medium used for the exosomes cultures in section 2.2
2. Mention which type of electron microscopy is used for the exosome structural analysis in line 152.
3. What are the major differences between CPC1-CPC3? In Fig 2D, why certain biomarkers are absent for CPC1-CPC3? Authors need to explain this in the results or discussion section.
4. Provide the abbreviations for the terms where it is initially used.
5. Correct the error from “Opti-MEN” to “Opti-MEM” in line 271
There are some font size and spelling error appears in the manuscript.
Author Response
Please find the reply as attached

Round 2
Reviewer 1 Report
The authors addressed all my questions.
Author Response
Thanks for your review.